# The Mercury Concentration in Spice Plants

Agnieszka Fischer * and Barbara Brodziak-Dopierała

Department of Toxicology and Bioanalysis, Faculty of Pharmaceutical Science, Medical University of Silesia, 30 Ostrogórska Str., 41-200 Sosnowiec, Poland
* Correspondence: afischer@sum.edu.pl

**Abstract:** Spice plants are popularly used as ingredients in food products. Promoting healthy eating, paying attention to the quality of products, means that organic and self-produced ingredients, whose origin and growing conditions are known, are gaining popularity. The study determined the concentration of mercury (Hg) in popular leafy spice plants: peppermint (*Mentha piperita*), common basil (*Ocimum basilicum*), lovage (*Levisticum officinale*) and parsley (*Petroselinum crispum*). Self-grown spices and ready-made commercial products were selected for the study. The Hg content in the test samples was determined by the AAS method (AMA 254, Altec, Praha, Czech Republic). The range of Hg content in the tested spice samples ranged from 1.20 to 17.35 µg/kg, on average 6.95 µgHg/kg. The highest concentration of Hg was recorded in the peppermint, 9.39 µg/kg. In plants grown independently, the concentration of Hg was statistically significantly higher than in commercial products purchased in a store. There were no differences in the concentration of Hg in organic and non-organic spices. Commercial spices defined by producers as organic products did not differ statistically significantly in the level of Hg content from non-organic products.

**Keywords:** spice; plant; mercury

## 1. Introduction

Spice plants and herbs are widely used all over the world to improve and diversify the taste of food and as digestive enhancers and they also have a healing effect [1,2]. Their natural ingredients reduce the risk of developing cardiovascular and neurodegenerative diseases, cancer, obesity and type 2 diabetes [3–6]. The presence of biologically active compounds with antibacterial and anti-inflammatory properties extends the shelf life of food, and antioxidants cause an antioxidant effect [3,4]. Promoting a healthy lifestyle and diet has caused a growing interest in herbs, medicinal plants and spices. Spice plants are grown under natural conditions, but environmental pollution, the use of pesticides and the method of storage and processing may contribute to the increased presence of undesirable compounds in the final product. Soil is the main source from which metals can enter plants.

Metals most often occur in soil as insoluble forms, therefore plants regulate their sorption by secreting compounds that change the pH and facilitate uptake by the root system [7–11]. The uptake can be apoplastic, regulated by passive diffusion mechanisms. The symplastic pathway based on active transport against a concentration gradient with the use of carriers or complexing compounds is used more often [7,12]. After reaching the root cells, complexes are formed with chelating compounds (organic acids, polyphenols, proteins, metallothioneins, pectins). The resulting complexes are accumulated in the cell wall or in vacuoles, and then transported to the plant shoot through the xylem vessels in leaves [7,13]. Toxic metals can also move through cation channels in the cell membrane [7,8].

Due to the similar structure and the uniform uptake of toxic and essential metals, plants have developed defense mechanisms to minimize negative effects. One of them is the avoidance process, consisting in limiting the absorption of heavy metals from the soil thanks to the production of substances that reduce sorption in the roots and precipitation of insoluble forms [9,14]. The second mechanism is tolerance consisting in the uptake of

heavy metals from the soil, then, by chelation with substances produced in the plant, fewer toxic compounds are created, which are deposited in the vacuoles [7,15].

Mercury (Hg) is a toxic metal that accumulates in plants and leads to biomagnification at higher trophic levels, which contributes to a direct threat to humans [16–19].

The developing industry, as well as the increase in the combustion of fossil fuels, is the reason for the increase in Hg emissions to the natural environment [20–22]. It is especially dangerous for plants that are able to absorb this element from soil contaminated with industrial waste and rainwater [16,23].

Recently, a significant increase in the share of spices and herbs in Polish trade has been observed. It is estimated that one consumer uses approx. 16–17 bags of dried spices annually. The average spice sales growth rate is 10%. The inclusion of herbs in the pyramid of healthy eating and physical activity contributed to the increased interest in spices. Herbs are recognized, among others, as a healthy replacement for table salt. In Poland, the sale of bag spices imported, mainly from Vietnam and Germany, is dominant. However, growing one's own fresh herbs in households is becoming more and more popular [24,25].

The aim of this study was to assess the mercury content in leaf spices from own crops and in ready-made commercial products purchased in stores. Four species of the most popular capsule herbs from different producers and the same spice plants grown in home gardens were selected for the study. Home-grown spices were grown in five different locations in three voivodeships: Śląskie, Małopolskie and Opolskie. The cultivation sites were varied, in both rural and urban areas. The conducted research was aimed at comparing the concentration of Hg in ready-made and cultivated spices. The obtained values were compared with the permissible standards.

## 2. Material and Methods

The study included 48 samples of plants commonly used as spices. The aerial parts of four plant species were selected for the study: peppermint (*Mentha piperita*), common basil (*Ocimum basilicum*), lovage (*Levisticum officinale*) and parsley (*Petroselinum crispum*). The tested plants were divided into 2 groups: self-cultivated and commercial products. Commercial products were purchased in stores in Poland (*n* = 28). They came from popular spice producers. Among the samples purchased in stores, there were also products marked as organic, with the manufacturer's annotation that it is a product with a certificate of organic production. The description on the packaging explains that "plants harvested by hand from the natural state in special, certified ecologically clean zones were used to prepare the product, or they were produced on organic farms. On organic farms, no fertilizers or chemical plant protection products are used". The remaining part of the test samples (n = 20) were spice plants grown by consumers themselves in home gardens. These plants came from 5 different locations in three voivodeships in southern Poland: Śląske (3 collection points), Opolskie (1 collection point), Małopolskie (1 collection point). The locations of the plant cultivation points with geographic coordinates are presented in Table 1. Four tested plant species were collected from each location.

**Table 1.** Location of spice growing points in southern Poland.

| Place | Voivodeship | Geographical Coordinates |
|---|---|---|
| Katowice | | 50.2752559 19.0776667 |
| Mysłowice | Śląskie | 50.1907033 19.1535399 |
| Jastrząb | | 50.6652231 19.1607083 |
| Płoki | Małopolskie | 50.2025604 19.5033717 |
| Jemielnica | Opolskie | 50.5445516 18.3431949 |

Jemielnica and Jastrząb are villages where there are no large sources of pollution, such as industrial plants. Płoki is a town located in the south of Poland, near a coal-fired power plant and a closed hard coal mine. Katowice and Mysłowice are the cities of the voivodeship Śląskie, located in an area with a high degree of urbanization, exposed to pollution from nearby industrial plants such as steel mills, mines and heat and power plants, as well as high intensity of communication.

Two commercial packages of spices from the same producer were used for the study. The average test sample for each spice was prepared from the products. The test samples were ground in a porcelain mortar. In the case of spice plants grown on their own, leaf samples were collected from each tested species. The plant material was air-dried at room temperature in an airy and shaded room. After drying, the plant material was ground in a porcelain mortar. From each prepared sample of spices, 3 independent averaged weights of about 50 mg (RADWAG analytical balance, Radom, Poland) were prepared and used for analysis.

The Hg content in the test samples was determined by the AAS method (mercury analyzer AMA 254, Altec, Czech Republic). Measurement conditions used were: wavelength 253.65 nm, carrier gas—oxygen ($O_2$ purity $\geq$ 99.5%), inlet pressure 200–250 kPa, times of the individual stages of the analysis were: 120, 140, 60 s. Before each measurement, the apparatus was purged with air and distilled water according to the analytical procedure [26].

The detection limit is 0.01 ngHg/g. The original factory calibration was still valid for the calibration of the instrument. In order to verify the correctness of the applied method, Hg determinations were used in the reference material (Mixed Polish Herbs INCT-MPH-2, Institute of Nuclear Chemistry and Technology Department of Analytical Chemistry, Warsaw, Poland). The result of the Hg determinations with six repetitions for the reference material was 0.0166 $\pm$ 0.0001 mg/kg with a recovery of 92.22%. The detection technique used covers the total amount of Hg, regardless of its form in the sample. The final Hg concentration in the tested spice sample was the arithmetic mean of 3 measurements. This result was statistically analyzed using Statistica 13.3 (Statsoft, Kraków, Poland). Distribution of variables was evaluated by the Shapiro–Wilk test and quantile–quantile plot. The interval data were expressed as a median (lower–upper quartiles). The non-parametric Mann–Whitney U test (for two samples) and Kruskal–Wallis test (for a greater number of samples) were used in order to compare the data. Statistical significance was set at a *p*-value below 0.05 and all tests were two-tailed [27].

## 3. Results

The Hg content in the tested spice samples ranged from 1.20 to 17.35 µg/kg, arithmetic mean 6.95 µgHg/kg, median 6.17 µgHg/kg. A complete statistical analysis of the Hg concentration in the tested spice samples is presented in Table 2.

**Table 2.** Hg concentration in the tested spice plants, µg/kg.

| Species | Kind of Species | *n* | Mean $\pm$ Standard Deviation | Median | Quartile Q$_1$–Q$_3$ | *p* |
|---|---|---|---|---|---|---|
| | All | 48 | 6.95 $\pm$ 3.70 | 6.17 | 4.58–8.83 | |
| | Basil | 15 | 6.19 $\pm$ 2.08 | 5.52 | 4.72–7.24 | |
| | Lovage | 11 | 6.02 $\pm$ 3.67 | 5.27 | 3.08–8.27 | 0.2968 |
| | Peppermint | 10 | 9.39 $\pm$ 5.83 | 9.58 | 2.40–14.18 | |
| | Parsley | 12 | 6.72 $\pm$ 2.39 | 6.40 | 5.83–7.64 | |
| | Commercial packages | 28 | 5.15 $\pm$ 2.58 | 4.99 | 3.23–6.54 | <0.001 |
| | Grown independently | 20 | 9.47 $\pm$ 3.58 | 8.70 | 6.38–11.55 | |
| Commercial packages | Non-organic | 24 | 5.20 $\pm$ 2.64 | 4.99 | 3.65–6.19 | 0.97381 |
| | Organic | 4 | 4.88 $\pm$ 2.56 | 4.91 | 2.69–7.07 | |

The Hg concentration in individual spice plant species (basil, lovage, peppermint and parsley) did not differ statistically significantly. The highest average concentration of Hg was found in peppermint (9.39 μg/kg) and it was about 3 μg/kg higher than in other species. The range of Hg concentration in the tested peppermint samples was 1.20–17.35 μg/kg, and the highest value of the standard deviation (5.83 μg/kg) among the herbs tested indicates a particular differentiation of the metal in this plant. In the case of other spices, the average Hg concentrations were very similar and amounted to 6.72 μg/kg for parsley, 6.19 μg/kg for basil, and the lowest for lovage—6.02 μg/kg—shown in Table 2.

The analysis of spice plants in terms of the origin of the product and their quality showed that there were statistically significant differences in the concentration of Hg. Comparison of the samples of spices bought in stores and grown on their own showed a lower concentration of Hg in commercial products—shown in Table 2. These differences were statistically significant ($p$ = 0.00003). Cultivated plants had the highest Hg content, followed by popular commercial products (non-organic) and organic commercial products— shown in Figure 1.

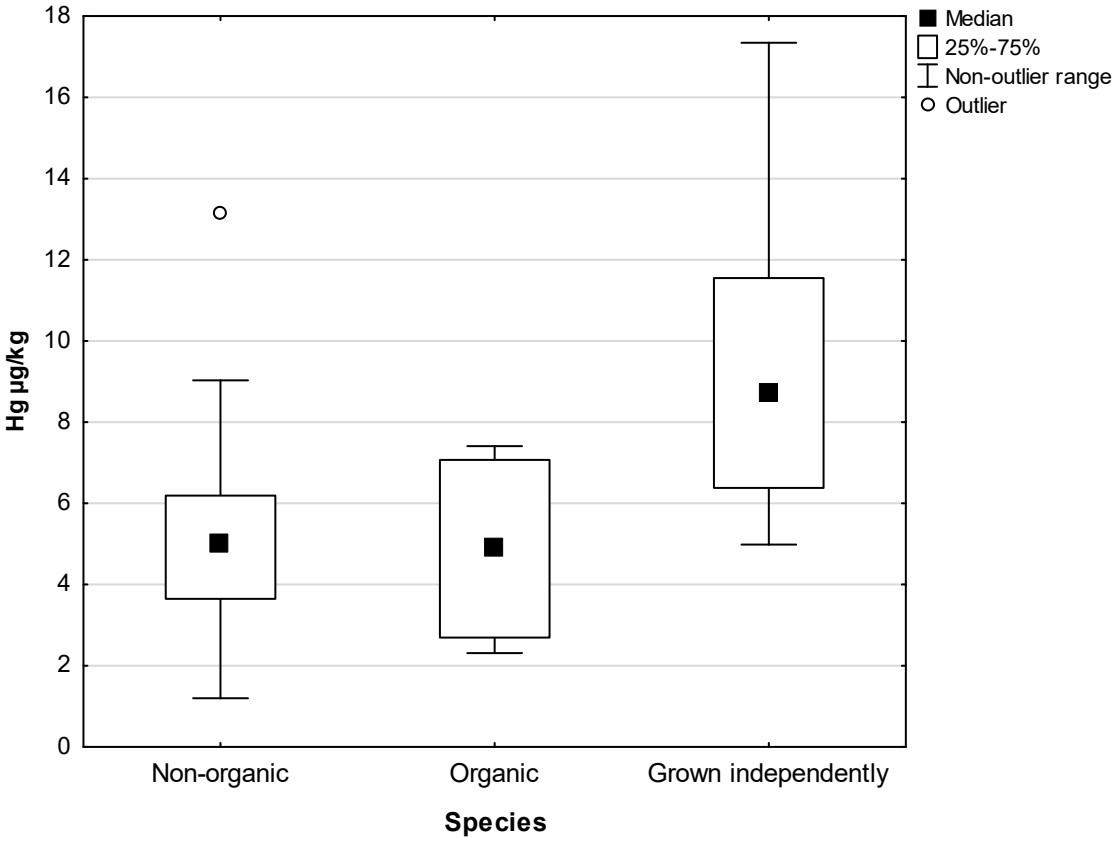

**Figure 1.** Hg concentration in organic and non-organic spices purchased in stores and grown independently, μg/kg.

The difference in the Hg content in organic and non-organic commercial products was not statistically significant. Among organic products, lower Hg concentrations were recorded in lovage and peppermint. In all analyzed commercial spices, the average Hg concentration was 5.15 μg/kg (Table 2), including for individual plants: parsley 6.03 μg/kg, basil 5.43 μg/kg, lovage 3.23 μg/kg, peppermint 2.40 μg/kg—shown in Figure 2. In spices from own crops, an average of 9.47 Hg μg/kg was found (Table 2), including: peppermint 14.18 μg/kg, parsley 7.62 μg/kg, basil 6.00 μg/kg, lovage 8.27 μg/kg—shown in Figure 2. Individual spice plants showed a higher concentration of Hg in all species grown independently—shown in Figure 2. However, statistically significant differences ($p$ < 0.05) were only found for lovage and peppermint.

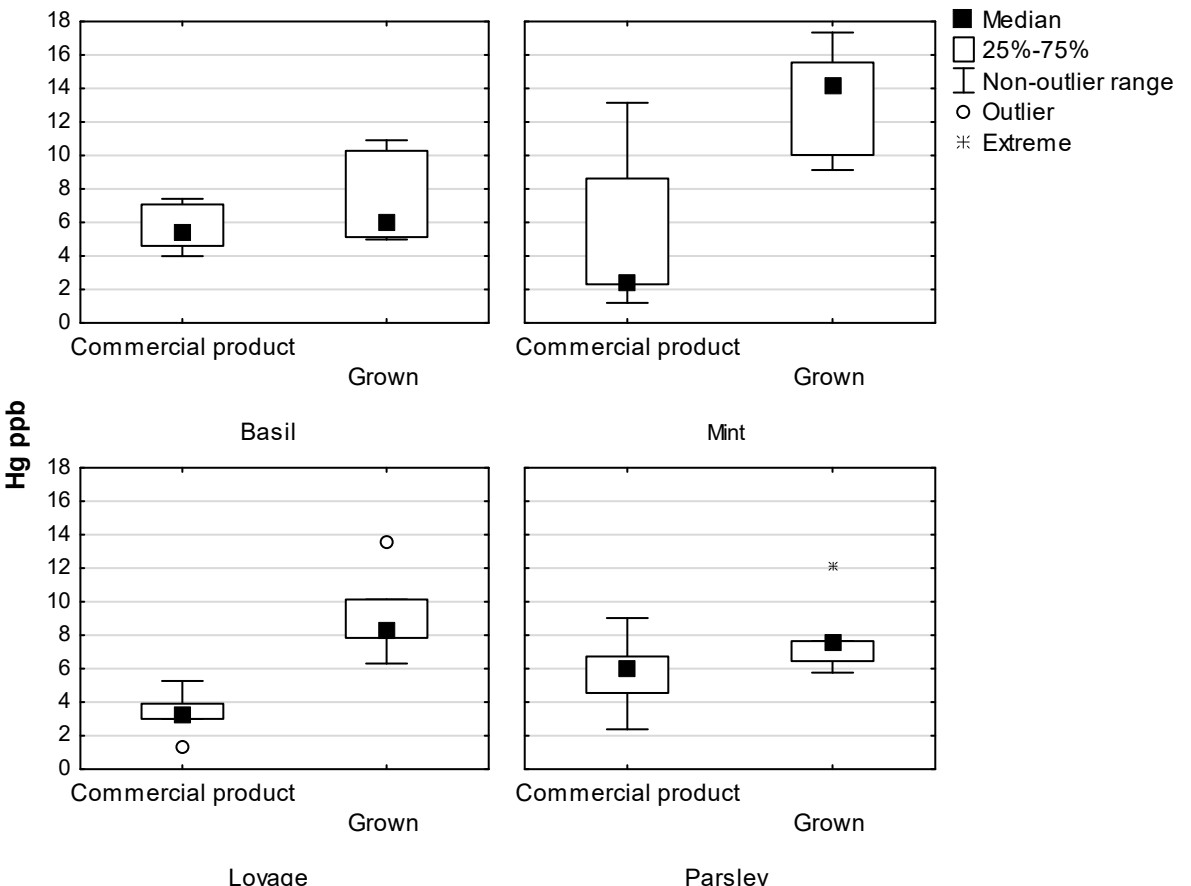

**Figure 2.** Hg concentration in study spice plants purchased in stores (commercial product) and grown independently, µg/kg.

On the other hand, the analysis of Hg concentration in spices grown independently in various locations (Śląskie, Małopolskie and Opolskie voivodeships) showed the highest concentration of Hg in plants from Małopolskie. In spices grown in Płoki (Małopolskie voivodeship), Hg concentration was the highest for all plants (12.90 µg/kg)—shown in Figure 3.

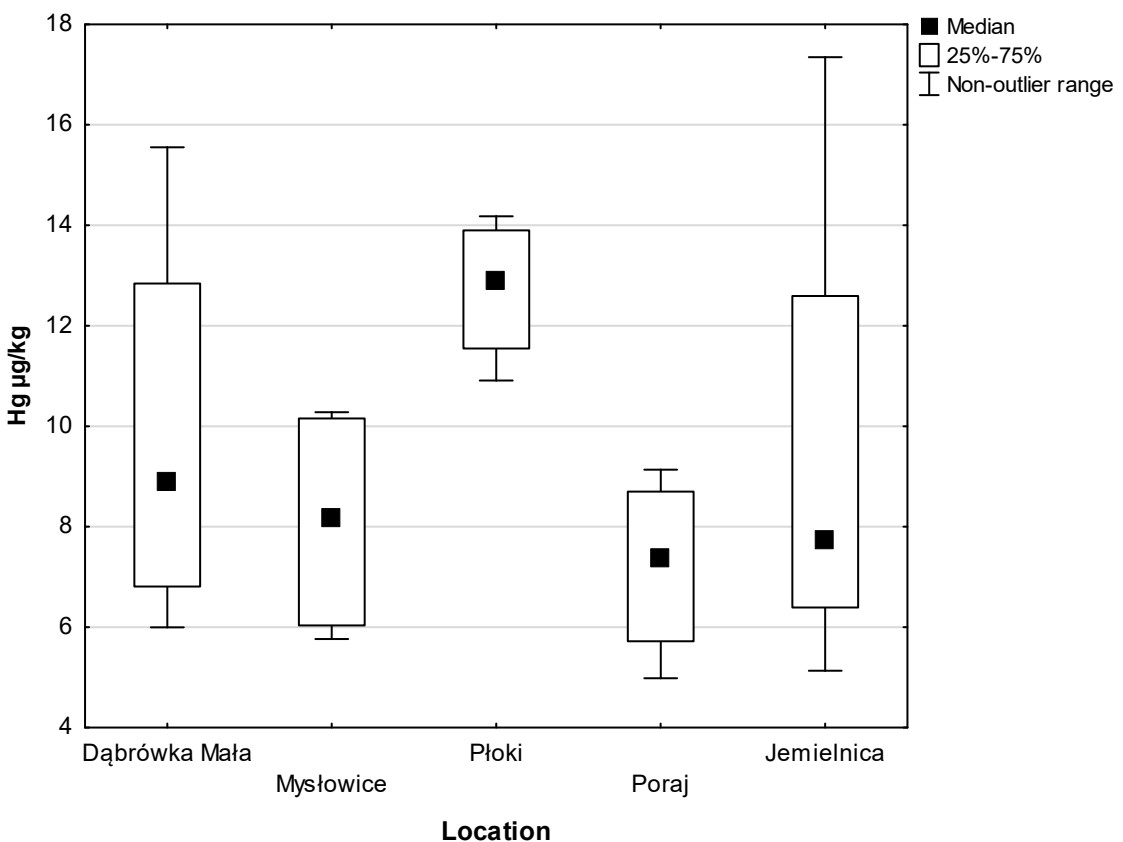

**Figure 3.** Hg concentration in spices grown independently in different locations in southern Poland, µg/kg.

## 4. Discussion

Spices are used not only for their organoleptic qualities, but also for their anti-inflammatory, antibacterial and antioxidant properties. In addition to beneficial substances, these plants may also contain toxic compounds, e.g., heavy metals, including Hg. For self-cultivated plants, the elements can permeate from soil, air and water. On the other hand, the sources of heavy metals in the case of commercial products may also be processing and storage processes.

The maximum level of mercury in deciduous spice plants (herbs) is set out in the Commission Regulation (EU) of 16 January 2018 amending Annexes II and III to Regulation 396/2005 of the European Parliament and of the Council with regard to the maximum residue levels of mercury compounds in certain products and on their surface. According to this regulation, the maximum level of mercury in leafy spice plants is 0.03 mg/kg [28]. Hg concentration in the tested herbs did not exceed the acceptable standard.

In the obtained research, attention is drawn to the differentiated content of Hg in spices from own farms compared to ready-made bag spices purchased in stores. The highest Hg content in spices grown independently concerns peppermint, then parsley, basil and lovage. The reverse sequence of Hg applies to ready-made spices and is as follows: parsley, basil, lovage, peppermint.

Literature data [29–31] confirm the increased ability to accumulate Hg in aerial parts of plants. Kwapuliński et al. [32], examining the dandelion, determined that the average Hg content in the stem was 5 µg/kg, in leaves 14 µg/kg and in roots 12 µg/kg. Guangle et al. [33] confirm the increased concentration of Hg in leaves (49 µg/kg) compared to corn seeds (10 µg/kg). Additionally, Jena and Gupta [30] indicate a higher concentration of mercury in basil leaves than in the rest of the plant. Similarly, Ordak et al. [31] notice significant differences in the mercury content between particular morphological parts of plants. Among the 20 different tested species of medicinal plants, the concentration of this

element in the leaves showed the highest range of 5.84–81.54 µg/kg [31]. In the leaves of the spice plants we examined, the results ranged from 1.20 to 17.35 µg/kg, including slightly more for our own crops of 4.98–17.35 µg/kg.

The reason for the increased concentration of mercury in the leaves is that, in addition to exposure from soil, the atmospheric air in the form of dry and wet deposition is an important source of contamination. Dust is deposited on the leaf blade, the amount of which depends on the weather conditions: the amount of rainfall, the direction and strength of the wind [31].

Chizzola [34] proves in his research that plants exhibit different, species-specific susceptibility to the influence of toxic substances, including heavy metals. In the authors' own research, no statistically significant differences were found in the concentration of Hg in individual species of herbs—peppermint, basil, lovage and parsley. On the other hand, statistically significant differences were observed when comparing samples of spices purchased in stores and grown independently.

Increased Hg content in self-cultivated herbs may result from growing conditions (soil type, mineral content, rainfall intensity, degree of contamination) and the specifics of a given species. Species variability in Hg concentration may result from the length of the growing season of individual plants, e.g., polycarpic plants show higher concentrations of this metal than monocarpic plants. This is confirmed by Driscoll et al. [35]—over time, as a result of accumulation, an increase in the level of Hg in long-term plants is noticeable.

Differences in Hg levels among the same plant species may be due to the season of harvesting. Ordak et al. [31] studied plants collected at different times. Their results showed that the Hg content in the same herb species was much higher in the fall than in the spring. These studies show that the Hg content in plants increases with the duration of the growing season. Herbs for our own research were collected at the same time (summer, June), about 2 months after germination, in the middle of the growing season. The amount of rainfall preceding the harvest is a factor that may also affect the determined amount of Hg in plants. Heavy rainfall increases exposure to metal in the form of wet deposition.

Baranowska et al. [36] showed a relationship between the concentration of Hg in herbs depending on the place of cultivation. Relatively high concentrations of heavy metals were determined in leaf spices grown near a railway station. These concentrations were comparable to samples from areas close to roads and highways. Our own research showed differences between mercury concentrations in different locations, which concerned three voivodeships in southern Poland. According to the average values obtained for plants in individual cultivation areas, the highest Hg content was recorded in spices from Płoki (Małopolskie voivodeship), and the lowest from Jemielnica (Opolskie voivodeship).

Research conducted in Saudi Arabia compared the mercury content in plants and vegetables from markets located in different cities of this country [37]. For example, Hg concentrations in parsley varied from 16 to 48 µg/kg depending on the place of purchase. The increased content of Hg and other heavy metals in the central district may be the result of heavy traffic and industrial development [37]. In our research, the Hg concentration in the cultivated parsley leaves was lower (5.76–12.19 µg/kg), while a higher metal content was found in plants grown in a highly urbanized area with a particularly high traffic intensity of Śląskie voivodeship (the large city - Katowice, and a smaller city – Mysłowice). The highest concentration of Hg in parsley, 12.19 µg/kg, also in other tested plants, was found in the crops in Płoki. The high concentrations of mercury in the leaves of spice plants from Płoki can be explained by the fact that there is a power plant in the vicinity of this location, which uses coal as fuel, which may contribute to the increased content of Hg in plants growing in this area.

Gibicar et al. [38] analyzed the concentrations of heavy metals in plants from the village of Rosignano Solvay in Tuscany (Italy), located near the chlorine-producing Solvey plant. The highest concentrations of Hg were recorded in peppermint (47.9 µg/kg), then in basil (32.5 µg/kg) and parsley (26.3 µg/kg). In the authors' own study, the mean concentrations of mercury in plants from cultivation were much lower and amounted to: peppermint

9.58 μg/kg, parsley 6.40 μg/kg, lovage 5.27 μg/kg and basil 5.52 μg/kg. In addition, Baranowska et al. [36], Nordin and Selamat [39], Ali and Al-Qahtani [37] and Can et al. [40] show that the content of Hg in cultivated plants depends on the degree of environmental pollution. Similarly, in the authors' own research, differences in the concentration of this element were noted between the places of cultivation. However, our results show that areas classified as rural, which are potentially low in pollution, have a high concentration of mercury in plants. For example, the concentration of mercury in herbs grown in the rural area of Jastrząb is very similar to that of cultivation in large cities (Katowice and Mysłowice) and lower than in the other studied rural area—Płoki.

Driscoll et al. [35], Ordak et al. [31], Guangle et al. [33] and Kwapuliński et al. [32] indicate in their works that the Hg content in deciduous plants depends on many other factors, such as: species, duration of the growing season, soil type and amount of precipitation.

In the scientific literature, there are no studies assessing the difference between the mercury content in the same species of ready-made (commercial) plants and those grown independently. On the basis of the results, a statistically significant difference was noted between leaf spices from producers and cultivated spices. Hg concentrations in peppermint, parsley, lovage and basil from individual locations were higher than those obtained by purchasing ready-made products. Food products, including spices, are subject to numerous quality and composition controls. According to a report by the Office of Competition and Consumer Protection (UOKiK) of 2019, among the 20 samples of spices available on the market, as many as nine of them showed inconsistencies in their composition, which accounted for 45% of the total [41]. Most often, they resulted from the lack of species purity, the presence of individual hairs and animal hair, falsification with other plants, e.g., in oregano, the presence of black and red currant leaves, marjoram and thyme was found. The addition of cheaper ingredients, such as woody stems, reduces the content of compounds responsible for the unique properties of individual spices. The decrease in the content of essential oils lowers the taste and aroma values, but also reduces the antioxidant and anti-inflammatory effects [41]. The above studies and the available scientific literature confirm the influence of the environment on the increase in mercury concentration in herbs.

The most recent research on the concentration of heavy metals in commercial spices in Poland was carried out by Kowalska [29]. The mercury values in the basil samples ranged from 5 to 30 μg/kg. In this spice, the Hg content was lower, although similar to the lower range of 4.72–7.24 μg/kg.

Kowalska [29] estimated the risk assessment of exposure to mercury in her research. She took into account in her research that the average daily consumption in Poland of spices and herbs is 0.7 g/day, and the amount of Hg consumed was 0.00003 mg/kg/day (mg trace element/kg body weight/day) [29]. Research shows that Hg contained in spices in the Polish diet is not a source of danger to humans. Taking into account the average concentration of Hg expressed in μg/kg, the average daily consumption of herbs and spices (0.7 g/day) and the average weight of an adult (70 kg), the estimated daily consumption (EDI) of the analyzed element was calculated. For the tested cultivated plants, the EDI value was 0.0000515 mg/kg/day, and in the group of commercial products it was 0.0000947 mg/kg/day. The value obtained for the herbs from the cultivation and from the producer exceeded the dose calculated by Kowalska [29].

Both the spices studied in this study and the aforementioned studies [29–38] indicate that the acceptable standards for heavy metal content were not exceeded. In comparison with the literature data [29–38], the Hg concentration values obtained in our studies were lower. Based on the results of this research, it can be concluded that the analyzed spices and herbs available on the Polish market are not dangerous to consumers and do not exceed the permissible levels of mercury. Our research included selected samples of spice plants and their limited numbers, which may constitute a study limitation. Due to the changing level of heavy metals in food products and their diversity, it is advisable to continue research on the content of Hg in spice plants.

## 5. Conclusions

- The Hg content in the tested spice plant samples ranged from 1.20 to 17.35 µg/kg. The highest average concentration of Hg was recorded in the samples of peppermint at 9.39 µg/kg, which was higher by about 3 µg/kg than in the other species of tested spices (parsley, lovage, basil).
- A statistically significantly higher concentration of Hg was found in plants grown independently than in commercial products.
- Commercial spices defined by producers as organic products did not differ statistically significantly in the level of Hg content from non-organic products.
- In the plants grown independently in the selected locations, the Hg concentration was varied. In rural areas, it was similar to or significantly higher than in metropolitan areas.
- Hg concentrations in all spice plant samples did not exceed the applicable legal standards.

**Author Contributions:** A.F.: conceptualization, methodology, validation, visualization, funding acquisition, writing—review and editing, supervision. B.B.-D.: conceptualization, investigation, formal analysis, writing—original draft preparation, writing—review and editing. All authors have read and agreed to the published version of the manuscript.

**Funding:** This research was funded by Medical University of Silesia, number PCV-1-176/N/1/I.

**Data Availability Statement:** The data presented in this study are available on request from the corresponding author.

**Conflicts of Interest:** The authors declare no conflict of interest. The sponsors had no role in the design, execution, interpretation or writing of the study.

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
