# Peer review of "The Mercury Concentration in Spice Plants"

_processes, doi:10.3390/pr10101954_

Round 1

Reviewer 1 Report

Please find in the following my comments about the review of a manuscript under the title (The mercury concentration in spice plants). In this study, the authors determined the concentration of mercury (Hg) in popular leafy spice plants

Main question addressed by the research

This study focused to investigate the concentration of mercury (Hg) in popular leafy spice plants. The aim of this study was to assess the mercury content in leaf spices from own crops and in ready-made commercial products purchased in stores

Originality and relevance

§  The study is interesting for reading and relevant in the field.

§  The study has moderate scientific quality.

§  The study is relevant to the scope of this journal.

§  The manuscript is clear, and relevant for the field and its presentation needs minor modifications.

Comments:

Results:

§  The title and legends of the table and figures should be informative and self-explanatory. Revise.

Conclusion:

§  The conclusion should be shortened, and the remaining data can be mentioned in the result and discussion section.

§  Remove the sentence (The analyzed leaf spices available on the Polish market do not pose a health risk to consumers in terms of Hg content.) as the authors did not study this part.

Reviewer 2 Report

The authors made an experimental study, measuring Hg content in 4 leafy spices, and comparing the content among the species, and among the commercial and self-grown spices. Measurement methodology is appropriate, number of specimens sufficient to provide acceptable statistical power. Data distribution was checked, and appropriate nonparametric tests used. Language of the article and writing style are acceptable. The manuscript merits acceptance, with one minor change:

1. a paragraph about the study limitations should be added at the end of the Discussion section

Reviewer 3 Report

This manuscript evaluates and compare the concentration of mercury (Hg) in Self-grown spices leafy spice plants and ready-made commercial products.  The 13 range of Hg content in the tested spice samples ranged from 1.20 to 17.35 μg/kg, on average 6.95 14 μgHg/kg. The work is well conducted, with appropriated technics. The results are convincing and seem trustable. However, the work it’s not original by itself because studies on mercury concentrations in plants has been already described in the literature.

Minor points

·       Authors should add a conclusion to the abstract

·       In the abstract line 10 the word “peppermint” should be rectified “in popular leafy spice plants: pepperpeppermint.

·       Figures quality should be improved  
